# Wound and Short-Term Scar Outcomes of Meek Micrografting Versus Mesh Grafting: An Intra-Patient Randomized Controlled Trial

**DOI:** 10.3390/ebj6020026

**Published:** 2025-05-19

**Authors:** Danielle Rijpma, Karel Claes, Anouk Pijpe, Henk Hoeksema, Ignace De Decker, Jozef Verbelen, Matthea Stoop, Kimberly De Mey, Febe Hoste, Paul van Zuijlen, Stan Monstrey, Annebeth Meij-de Vries

**Affiliations:** 1Alliance of Dutch Burn Care, Burn Center Beverwijk, Red Cross Hospital, Vondellaan 13, 1942 LE Beverwijk, The Netherlands; apijpe@rkz.nl (A.P.); mstoop@rkz.nl (M.S.); ppmvanzuijlen@brandwondenzorg.nl (P.v.Z.); adevries@rkz.nl (A.M.-d.V.); 2Department of Plastic, Reconstructive and Hand Surgery, Vrije Universiteit Amsterdam, 1081 HV Amsterdam, The Netherlands; 3Amsterdam Movement Sciences, Tissue Function and Regeneration, Boelelaan 1117, 1081 HV Amsterdam, The Netherlands; 4Department of Plastic Surgery, Ghent University Hospital, C. Heymanslaan 10, 9000 Ghent, Belgium; karel.claes@uzgent.be (K.C.); hendrik.hoeksema@uzgent.be (H.H.); ignace.dedecker@ugent.be (I.D.D.); stan.monstrey@uzgent.be (S.M.); 5Burn Center, Ghent University Hospital, C. Heymanslaan 10, 9000 Ghent, Belgium; jozef.verbelen@uzgent.be (J.V.); kim.demey@uzgent.be (K.D.M.); 6Faculty of Medicine and Health Sciences, Ghent University Hospital, 9000 Ghent, Belgium; febe.hoste@ugent.be; 7Department of Plastic, Reconstructive and Hand Surgery, Red Cross Hospital, Vondellaan 13, 1942 LE Beverwijk, The Netherlands; 8Department of Pediatric Surgery, Amsterdam UMC, Location AMC, Meibergdreef 9, 1105 AZ Amsterdam, The Netherlands; 9Department of Surgery, Red Cross Hospital, Vondellaan 13, 1942 LE Beverwijk, The Netherlands

**Keywords:** burn wounds, scars, randomized controlled trial, mesh grafting, meek micrografting

## Abstract

Mesh grafting and Meek micrografting are split-thickness skin graft expansion techniques. This study aimed to compare the effectiveness of Meek and Mesh expansion ratios 1:2 and 1:3 in smaller wounds. An intra-patient randomized controlled trial was conducted at two burn centers (the Netherlands and Belgium). Wound outcomes, e.g., take rate, re-epithelialization rate, and donor site size, were measured. At 3 months post-surgery, patient preference and scar quality were evaluated with the Patient and Observer Scar Assessment Scale (POSAS), cutometer and dermaspectrometer. Seventy patients with a TBSA of 10 ± 10% (mean ± SD) were included. The take rate was 79 ± 25% vs. 87 ± 19% (*p* = 0.003), Meek vs. Mesh, respectively. At follow-up, a majority of observer and patient POSAS items were statistically significantly lower, corresponding with better scar quality for Mesh grafting compared to Meek micrografting. The scar elasticity was 0.37 ± 0.20 vs. 0.42 ± 0.21 (*p* = 0.013) and mean melanin 13.3 ± 8.3 vs. 12.1 ± 7.7 (*p* = 0.019) for Meek vs. Mesh, respectively, and the patient preference was 32%, 49%, and 19% for Meek, Mesh, and no preference. Other outcomes showed no statistically significant difference. In patients with smaller wounds, Mesh showed superiority on most wound and short-term scar results. Nevertheless, patient preference within the 1:3 expansion ratio group and donor site size were in favor of Meek.

## 1. Introduction

The surgical treatment of burn wounds or comparable skin defects often consists of debridement, followed by reconstruction with an autologous skin graft, mostly a split-thickness skin graft (STSG). To minimize donor site morbidity while achieving optimal wound coverage, STSGs are often expanded. The most commonly used technique is Mesh grafting, which is easy and quick to perform. In practice, STSG expansion rates of 1:1 to 1:3 are mostly used [1]. Due to these relatively small expansion ratios, Mesh grafting is less suitable for the reconstruction of extensive wounds. Meek micrografting, on the other hand, is an STSG expansion technique initially designed for extensive burn wounds with associated higher expansion ratios of 1:3 to 1:9 [2].

However, a recent development of this technique allowed for a reduction of the minimum expansion ratio from 1:3 to 1:2, making it more suitable for reconstruction of smaller wounds. Both expansion techniques now enable a 1:2 expansion ratio, which results in an overlap in surgical indications for both techniques. Moreover, previous studies have shown that the actual expansion ratios of STSGs do not reach the initiated expansion ratios [3,4,5,6,7,8]. For example, Mesh grafts with an initiated expansion ratio of 1:3 correspond to an actual expansion ratio of 1:1.5 [5]. The actual expansion ratios of Meek micrografts seem to correspond more to the initiated expansion ratios than Mesh grafting, given that 85.5% to 99.8% of the initiated expansion ratios of Meek micrografts were achieved in the study of Lumenta et al. [6]. Nevertheless, there are no studies on the actual expansion ratio of Meek micrografting 1:2.

In the past, several studies compared Meek micrografting and Mesh grafting techniques [9,10,11,12,13]. However, an intra-patient comparison of the effectiveness of Meek micrografting and Mesh grafting in smaller wounds is currently lacking [14]. Therefore, we initiated an intra-patient randomized controlled trial (RCT) comparing Meek micrografting versus Mesh grafting with the same expansion ratio in burn wounds or comparable skin defects that require a 1:2 or 1:3 expansion STSG. The assessment of scar quality at the 12-month follow-up was the primary outcome parameter [15].

## 2. Materials and Methods

### 2.1. Study Design

This study was a multicenter, intra-patient, randomized controlled trial, conducted in the burn centers of the Red Cross Hospital at Beverwijk, the Netherlands, and Ghent University Hospital, Belgium. The study was approved by the Medical Ethics Committees of the VU medical center Amsterdam (NL74274.029.20) and Ghent University Hospital (B670201942116), and by the Institutional Review Boards of both hospitals on 22 June 2021. The study was carried out in accordance with the guidelines set forth in the Declaration of Helsinki (64th WMA General Assembly, Fortaleza, Brazil, October 2013), the General Data Protection Regulation, and the principles of Good Clinical Practice. The study protocol was registered as NL8847 and on the internet portal of the Central Committee on Research Involving Human Subjects (NL74274.029.20). The complete study protocol was published elsewhere [15].

### 2.2. Study Population

Table 1 shows the eligibility criteria. Patients aged 18 years and older with burn wounds or other deep skin defects requiring debridement and skin grafting underwent screening for enrollment in the study. Furthermore, eligible patients had to have two comparable wound sections in terms of depth, ideally confirmed using laser Doppler imaging (moorLDI2-BI, Moor Instruments Ltd., Axminster, UK), and each study area should have a minimum surface area of 36 cm^2^. Exclusion criteria were participation in another intervention study within the last 30 days; expected non-compliance due to cognitive impairment; and wounds only covering the face, hands, or joints were excluded because these locations are covered with STSGs with a maximum expansion ratio of 1:1.5, which is not achievable with Meek micrografting, and are thus unsuitable as study areas.

Patients were recruited in the intensive care units, wards, and outpatient clinics of both burn centers. Both oral and written study information was provided. If patients (or legal representatives) chose to participate, written informed consent was obtained.

### 2.3. Assignment and Randomization of Study Areas

Before surgery, two wound areas (study areas A and B), which matched in terms of depth, location, and size, were designated within the same patient. Per-operatively, after achieving hemostasis, the comparability of the wound areas was re-assessed, and areas A and B were defined. Next, study area A was randomly assigned to a skin expansion technique: Meek micrografting or Mesh grafting, using the REDCap randomization module (REDCap 11.1.19—© 2022 Vanderbilt University, Nashville, TN, USA). Study area A was grafted first, followed by study area B.

### 2.4. Treatment

An expansion ratio of 1:2 was used for wounds requiring skin grafting that covered <11% of the affected total body surface area (TBSA), while 11–20% of the affected TBSA was subjected to an expansion ratio of 1:3. Wound swabs were collected from both study areas, followed by wound debridement. The same debridement technique and expansion ratios were used for both study areas. Moreover, the study areas were subjected to the same wound dressings based on wound cultures and according to local standard-of-care of the burn centers. If this resulted in a different treatment between the study areas, this was noted in the CRF.

#### 2.4.1. Mesh Graft Technique

The size of the study area was calculated and used to estimate the size of the STSG. An STSG was harvested on the location preferred by the patient using a Zimmer^®^ dermatome (Zimmer Biomet Foundation Inc., Warsaw, IN, USA). The STSG was passed through the Meshing device, using the appropriate carrier for the required expansion ratio. The Meshed graft was positioned on the debrided wound area, trimmed, and fixed with Urgotul^®^/Surfasoft^®^ and staplers. Gauzes soaked in antimicrobial solutions based on wound cultures, followed by dry sterile gauzes and bandages, were placed over the Meshed grafts.

#### 2.4.2. Meek Micrograft Technique

The procedure of determination of location and size of donor site, as well as the harvesting of the STSG was as described for Mesh grafting. The STSG was cut into squares using a Meek micrograft cutting machine (Humeca, Borne, The Netherlands). These squares were attached to a pre-folded dual gauze called a plissé, which determines the expansion ratio. The plissé was applied to the wound and fixed with staples. The coverage of plissés was similar to the Mesh graft technique. Ideally, the required two STSGs were harvested as one contiguous donor site. Alternatively, two separate donor sites were harvested adjacent from each other. Figure 1 shows illustrations and Figure 2 photos of the study design: STSGs expanded by the Mesh graft technique and the Meek micrograft technique. For a detailed description of both Meek and Mesh grafting techniques, including step-by-step procedural guidance, please see the study protocol [15].

### 2.5. Outcome Parameters

Multiple outcome parameters, assessed by the means of both objective and subjective measurement tools, were collected pre-, per- and post-operative, at the 3- and 12-month post-operative follow-up. This paper covers wound healing and scar quality at outcomes at the 3-month follow-up; other outcomes such as long-term scar quality and cost-effectiveness will be discussed in future papers.

#### 2.5.1. Per- and Post-Operative Wound Outcome Parameters

The size of the study area and donor site size were calculated with a 3D camera (inSight 3D wound measurement, eKare, Fairfax, VA, USA). The actual expansion ratio was calculated by the ratio of this study area size and donor site size. After 8 ± 2 days, the graft take of the study areas was assessed by the experienced clinical team (burn physicians, surgeons, plastic surgeons and burn care coordinators) and expressed in percentage. On post-operative days 14 ± 2 and 21 ± 2, the re-epithelialization percentage of both study areas and donor sites was evaluated. Moreover, the days to complete wound healing of study areas and donor sites were assessed and defined as ‘the number of days post-operative at which the wound is re-epithelialized for >95%, has no more drainage, and no longer needs substantial wound dressing’. This was usually assessed at outpatient clinic visits, where patients were not seen on a daily basis. Therefore, the time to complete wound healing could be overestimated by a couple of days. Finally, patient preference and satisfaction with both study areas were assessed at 8, 14, and 21 ± 2 days. Preference was measured with the following question: “If you would have a new wound that requires skin grafting, which area would you prefer, considering both donor site sizes? Study area A, B, or no preference”. Satisfaction with both study areas was measured by a 5-point Likert scale (1 = very satisfied, 2 = a little satisfied, 3 = neutral, 4 = little dissatisfied, 5 = very dissatisfied).

#### 2.5.2. Short-Term Scar Outcome Parameters

At 3-months post operation, multiple scar outcome parameters were measured. Patients assessed the scar quality of both study areas in two ways. First, the patient scale of the POSAS 2.0 was used, a scar assessment scale consisting of six items covering visual, tactile and sensory aspects, and one overall opinion item, all scored on a 10-point scale with 1 representing normal skin and 10 representing the worst possible scar [16,17]. Second, patients were asked again their technique preference and satisfaction rate.

The researchers assessed the scar quality with three measurement tools. First, the scar quality was assessed with the observer scale of the POSAS 2.0 that contains visual and tactile aspects and one overall opinion [16,17] (see Appendix A). This assessment was performed by two clinicians (at least one researcher), and afterwards their scores were averaged to one mean score per item. Second, scar pliability was assessed with the cutometer (MPA 580, Courage + Khazaka electronic GmbH, Cologne, Germany) by measuring the maximal extension (Uf) and elasticity (Ue) of the scar [18]. Third, scar color was assessed with a dermaspectrometer (CORTEX Technology, Hadsund, Denmark), which measured the erythema and melanin values of the scar and normal skin [19]. For cutometer and dermaspectrometer assessments, five measurements within each scar were performed and averaged afterward. Moreover, the number of re-operations in the study areas within the follow-up period of 3 months was evaluated. Finally, subgroup analyses were performed on specific outcome parameters based on the main results.

### 2.6. Statistical Analysis

Descriptive statistics were performed for the patient, wound, and outcome parameters. Next, the differences in outcome parameters were statistically tested. A paired *t*-test or Wilcoxon signed-rank test was used for normally and non-normally distributed outcomes, respectively. Chi-squared tests were used for categorical outcome parameters. Analyses were conducted using SPSS PASW Statistics 25.0 (IBM, New York City, NY, USA) version, and a *p*-value of <0.05 was considered a statistically significant difference. For power analysis, see the protocol publication. In brief, a sample size of 70 patients yields 90% power when assuming a true mean difference of 1, a standard deviation of 2.4, a moderate correlation of 0.5, and a drop-out rate of 10% [15].

## 3. Results

In total, 566 patients were screened and 77 patients were included in a period from July 2021 until August 2023. After inclusion, seven patients appeared not eligible for randomization; therefore, the intervention was performed in 70 patients (see Figure 3). Patient and wound characteristics are shown in Table 2. For a photo series of a study patient before debridement, after skin grafting, and at follow-up measurements, see Figure 2.

### 3.1. Per-Operative Wound Results

Table 3 shows per- and post-operative wound results. The size of the study wounds was similar, but the donor site size of Mesh grafting was significantly larger. In addition, the actual expansion ratio of Mesh grafts was lower compared to both the initiated expansion ratio of Mesh grafts and the actual expansion ratio of Meek micrografts.

### 3.2. Post-Operative Wound Results

The take rate of the Meek study area was 79 ± 25% lower compared to the 87 ± 19% take rate of the Mesh study area (*p* = 0.003). Re-epithelialization showed comparable results: the Meek study area had a lower rate of 80 ± 22 compared to 90 ± 13% of the Mesh study area (*p* = 0.001).

At hospital discharge, 30 patients reported their preference for Meek or Mesh, which was almost equally divided: 11 patients preferred Meek, 12 patients preferred Mesh, and 7 patients had no preference (see Table 4, Table 5 and Table 6). This shows satisfaction for both techniques. Patients were a little or very satisfied 20/29 (69%) with both Meek and Mesh. The most remarkable difference was found in dissatisfaction, with 3/29 (10%) of the patients a little or very dissatisfied with Meek micrografting compared to 1/29 (3%) for Mesh grafting. Overall, the satisfaction levels for both techniques were also fairly evenly divided, and the proportion between the techniques was also comparable.

In the period up to the 3-month follow-up, 14/70 (20%) patients were re-operated in one or both study areas. The majority of re-operated patients, 8/14 (57%), were re-autografted at the Meek study area. The mean Meek take rate of these eight patients was 56 ± 31%, which is much lower compared to the mean Meek take rate of 79 ± 25% of the total study population. This accounted for the decision to re-graft; no cases of late Meek graft necrosis were observed. In 6/14 (43%) patients, both study areas were involved and no patients were re-operated for the Mesh study area only.

### 3.3. Short-Term Scar Results

#### 3.3.1. Patients

At the 3-month follow-up, patients rated all POSAS items higher for the scar in the area transplanted with Meek (‘Meek-scar’), corresponding with a worse scar quality compared to the area transplanted with Mesh (‘Mesh-scar’). This was significant for the items ‘relief’, ‘color’, ‘overall opinion’, and ‘thickness’, with differences in scores of 1.1, 1.0, 0.8, and 0.6 points, respectively. For all POSAS results, see Table 7. Moreover, 32% preferred the Meek, 49% of patients declared to prefer the Mesh, and 19% had no preference. The patient technique preference shifted more to Mesh grafting compared to the technique preference assessed at hospital discharge: Meek vs. Mesh, respectively, 37% vs. 40%; 32% vs. 49% (see Table 4). The satisfaction levels of both techniques were quite similar to each other; almost three quarters of the patients were satisfied with both techniques. Although the proportions were different between both techniques, patients were more often ‘very satisfied’ and were never ‘very dissatisfied’ with Mesh grafting compared to Meek micrografting (see Table 5 and Table 6).

#### 3.3.2. Observer

Clinicians also rated higher POSAS scores on all items for the Meek study area, which were significant for all items except for pigmentation (see Table 7). The mean scar pliability measured with the cutometer showed a significantly 0.05 points lower elasticity (Ue) score for Meek compared to Mesh (see Table 8). In addition, the mean melanin score measured with the dermaspectrometer was significantly higher in the Meek study areas compared to the Mesh study areas. The mean erythema score did not differ between both study areas (see Table 9).

### 3.4. Subgroup Analysis

#### 3.4.1. Delayed Wound Healing and Wound Colonization

Delayed wound healing in both study areas, defined as wound healing > 21 days, occurred in 38/70 (54%) patients. Delayed wound healing was observed for the Meek study area alone in 12 out of 70 (17%) patients, while no delayed wound healing was observed in the Mesh study area alone. Per-operative wound swab results were missing for 6/70 patients of the total study population, leaving the wound swab results of 64 patients for analysis.

A separate analysis was performed with the subgroup in which only the Meek study area showed a delayed wound healing. In this subgroup, the wound swab results of 1/12 patients were missing, leaving the wound swab results of 11 patients for analysis. In 9/11 (82%) patients, non-pathogenic flora or negative per-operative wound swab results were observed for both study areas, compared to 48/64 (75%) patients in the total study population. In 2/11 (17%) patients, both study areas were positive for *S. aureus*, compared to 9/64 (15%) patients in the total study population. In 1/12 (8%) patients, the Meek study area showed non-pathogenic flora, while the Mesh study area was positive for *S. aureus*, compared to 1/70 (1%) patients in the total study population, which was the same patient as in the subgroup. No per-operative wound swabs were found positive for *P. aeruginosa*; however, at post-operative day 8, *P. aeruginosa* was found in both study areas of 1/11 (9%) patients, compared to 2/64 (3%) patients in the total study population. Table 10, Table 11, Table 12 and Table 13 shows the wound colonization results for both the total study population and subgroups.

Additionally, we identified 15/70 (21%) patients who had a difference in complete wound healing time between the study areas of more than 7 days. Among this subgroup, wound swab results of 1/15 (7%) patients were missing, leaving the wound swab results of 14 patients for analysis. Moreover, 12/14 (86%) patients had non-pathogenic flora or negative wound swab results for both study areas. In 2/14 (14%) patients, both study areas were positive for *S. aureus*, no patients showed positive wound swab results for *P. aeruginosa*.

#### 3.4.2. Delayed Wound Healing and Preference of Technique

The proportion of the patient preference technique for Meek, Mesh, and no preference at the 3-month follow-up was, respectively, 19/59 (32%), 29/59 (49%) and 11/59 (19%). In patients with an uneventful wound healing time (≤21 days) for both study areas (n = 17), the technique preference was 5/17 (29%), 10/17 (59%), and 2/17 (12%), respectively, Meek, Mesh, and no preference. If the wound healing of both study areas was delayed (n = 32 patient preference scores of 6/38 patients were missing), the technique preference was 12/32 (38%), 12/32 (38%), and 8/32 (25%), respectively, Meek, Mesh, and no preference. In the case of the delayed wound healing of the Meek study area only (n = 10, patient preference scores of 2/12 patients were missing), the technique preference shifted to 2/10 (20%), 7/10 (70%), and 1/10 (10%), respectively, Meek, Mesh, and no preference. Delayed wound healing exclusively within the Mesh study area did not occur.

#### 3.4.3. Expansion Ratio

The potential influence of the expansion ratio was studied on the outcome parameters. The graft take, time to complete wound healing, and scar quality measured with POSAS, cutometer, and dermaspectrometer results did not significantly differ between the two expansion ratio groups, except for the outcome patient preference at the 3-month follow-up, with 19/59 (32%), 29/59 (49%), and 11/59 (19%) in the 1:2 subgroup to 6/9 (67%), 2/9 (22%), and 1/9 (11%) in the 1:3 subgroup for Meek vs. Mesh vs. no preference, respectively.

## 4. Discussion

Mesh grafting showed superiority for most outcomes for wound results: the take rate, re-epithelialization rate, time to complete wound healing, and for scar results at the 3-month follow-up; patient preference and satisfaction rates, scar elasticity by the cutometer, pigmentation scores compared with normal skin by dermaspectrometer and most scar quality items by the POSAS.

The differences in patient POSAS items at the 3-month follow-up, including ‘color’, ‘thickness’, ‘relief’, and ‘overall opinion’ between Meek micrografting and Mesh grafting, were all significantly in favor of Mesh grafting, ranging from 0.6 to 1.1 points. For the observer POSAS, the items ‘vascularity’, ‘thickness’, ‘relief’, ‘pliability’, ‘surface area’, and ‘overall opinion’ were significantly lower for Mesh grafting compared to Meek micrografting, ranging from 0.3 to 0.8 points. Legemate et al. (2024) demonstrated that the Minimal Clinical Important Difference (MCID) of the POSAS 2.0 was 0.39 points, meaning that the discovered differences of almost all POSAS items in this study are clinically meaningful [20]. In addition, the mean scar elasticity by the cutometer and mean melanin scores compared with normal skin by the dermaspectrometer showed a slight significant difference in favor of Mesh grafting. No significant differences were found for the patient POSAS items ‘pain’, ‘itch’, and ‘pliability’ and observer POSAS item ‘pigmentation’. Moreover, mean scarmaximal extension by the cutometer and erythema scores compared with normal skin by the dermaspectrometer showed no significant difference. Bagheri et al. (2024) [21] showed that the subjective and objective assessments of scar pigmentation could slightly differ. In this study, deviated scar pigmentation, compared to healthy skin, was more often observed with the subjective measurement tool (observer scale of the POSAS) compared to the objective measurement tool (mexameter). Therefore, the outcomes of objective (dermaspectrometer) and subjective (observer POSAS) measurement tools do not completely correlate with one other.

Two outcome parameters were in favor of Meek micrografting; the technique preference of patients in the 1:3 expansion group at 3-month follow-up and the donor site size. First, contrary to the 1:2 expansion ratio group and the total study population, the majority of the patients in the 1:3 ratio expansion group preferred Meek micrografting. Yet, this shift to the superiority of Meek micrografting is not consistent with other outcome parameters of the 1:3 expansion ratio group. The sample size of the 1:3 expansion ratio group (n = 10 patients) compared to the 1:2 expansion ratio group (n = 60 patients) could be a possible explanation. Second, the donor site size of Meek micrografting was statistically significantly smaller compared to Mesh grafting. However, this did not appear to play a crucial role in patients’ technique preference and satisfaction, since patients preferred and were more satisfied with Mesh grafting, taking the donor site size into account. This raises the question if donor site size matters in relatively smaller wounds with corresponding smaller donor sites. A previous study evaluated the long-term donor site satisfaction among patients and showed patients were the least satisfied with size and appearance, but were still satisfied to very satisfied with their donor sites [22]. Moreover, a smaller donor site size corresponds with a relatively larger expansion ratio and therefore relatively less grafted skin. Although the same initiated expansion ratio was used for Meek micrografts and Mesh grafts, the actual expansion ratios of Mesh grafts were significantly lower compared to those of Meek micrografts, meaning more skin is grafted to the wound area with the Mesh technique. Therefore, it is debatable if the comparison of the same initiated expansion ratio for Meek micrografting and Mesh grafting is correct. On one hand, this variation in the amount of grafted skin could have influenced outcome parameters such as re-epithelialization and scar quality. Previous studies on expansion ratios of Mesh grafts and Meek micrografts suggested the actual expansion ratios of both techniques do not match, but the extent of the discrepancy varies between these studies [4,6,8]. On the other hand, the take rate is not affected by the expansion, though it was also significantly lower in Meek micrografting compared to Mesh grafting. In addition, this is the first intra-patient randomized controlled trial that compares Meek micrografting and Mesh grafting, which justifies the choice to start with an equal expansion ratio comparison. Finally, It should be noted that our cohort comprised relatively small burn areas; consequently, certain advantages of Meek micrografting, such as true graft size and the possibility of higher expansion ratios, enabling a smaller donor site, may be less apparent in small grafting areas but are likely to emerge more clearly in larger burns. Two other studies make a clinical comparison of Meek micrografting and Mesh grafting with several aspects that resemble our study design [12,13]. The first study by Dahmardehei et al. (2020) evaluated the wound results of Meek micrografting and Mesh grafting in 20 patients with an intra-patient comparison, by a case-control design [12]. However, compared to our study larger expansion ratios of 1:4 and 1:6 were used and only mean expansion ratios of 5.7 ± 1.9 for Meek and 4.2 ± 1.8 for Mesh were reported; they did not describe whether the same expansion ratio was used for both areas. In addition, the mean percentage TBSA burned was 36.9 ± 16.6%, mean percentage affected TBSA covered with Meek micrografts was 39% and Mesh grafts was 30%, meaning relatively extensive burns were included. Re-epithelialization time and operation duration were significantly shorter for Meek micrografts than Mesh grafts. Therefore, the authors recommend using Meek micrografting in high surface area burns, since this is based on extensive burns with larger expansion ratios it could not be translated to our study population with relatively smaller wounds and expansion ratios. The second study by Noureldin et al. (2022) compared wound and short-term scar outcomes of Meek micrografting and Mesh grafting in pediatric patients, by a randomized inter-patient comparison [13]. Forty pediatric patients with a mean percentage TBSA of 18.2% were divided into two groups and received Meek micrografting or Mesh grafting both with an expansion ratio of 1:3. The take rate was significantly better in the Meek micrografting group. Re-epithelialization time was shorter (27 vs. 35 days) and wound infection occurred less (25% vs. 40%) in the Meek micrografting group compared to the Mesh grafting group, but both results were not significant. These wound results were contradictory to the wound results we found in our study. However, at 3-month follow-up scar quality measured with the POSAS showed Mesh grafting had significantly lower mean overall opinion scores of 1 and 1.2 compared to Meek grafting, respectively for patients and observers, corresponding with a better scar quality of Mesh grafting. These scar results were in line with the scar results found in our study. Nonetheless, this study was performed in children, while our study was performed in adults.

The strengths of this study are the design, an international multicenter intra-patient randomized controlled trial. Since both techniques were applied to the same patients, the differences in patient factors influencing wound healing and scar formation were eliminated. Moreover, this study evaluated multiple clinical-, patient-, and clinician reported outcome measurements with multiple measurement tools at both short- and long-term follow-up. For instance, scar quality was measured by both patients and clinicians with both subjective and objective tools resulting in a comprehensive overview of the outcomes of Meek micrografting and Mesh grafting in burn wounds or comparable skin defects. On the other hand, there are also some limitations of this study. We compared the predetermined 1:2 and 1:3 expansion ratios for both techniques. Due to the relatively smaller expansion ratio for Mesh grafting which is known from previous literature and confirmed in this study, the comparison between Meek micrografting and Mesh grafting might not be completely appropriate. Moreover, the donor site size was estimated based on the study area size. Nevertheless, estimation was performed by experienced (plastic) surgeons and burn physicians, in some cases too much or too little STSG was harvested. In case of too little STSG, extra STSG was harvested and the surface area was calculated and added up to the initial donor site surface area. However, in case of too much STSG, it was difficult to quantify the exact surface area of the used STSG. The measured donor site size was used to calculate the actual expansion ratio for Meek and Mesh grafts and therefore these expansion ratios might not be completely accurate. In addition, outcome parameters such as take rate and re-epithelialization rate were estimated by clinicians, instead of a measurement tool such as digital image analysis. However, previous studies have shown that clinical evaluation of take rate and re-epithelialization rate by experienced observers obtains reliable results [23,24]. A limitation of the present study is that ten patients were lost to follow-up, exceeding the seven patients originally anticipated. Furthermore, patient preferences at the time of hospital discharge were not consistently recorded. This inconsistency arose for two reasons: first, a substantial proportion of patients received care on an outpatient (day treatment) basis; and second, several patients were inadvertently overlooked at the moment of discharge

Our future publications will reveal if the outcomes at the 12-month follow-up stay in favor of Mesh grafting and will present a cost effectiveness analysis to make a comprehensive assessment of the use of both techniques in relatively smaller wounds. Finally, a suggestion for future studies would be to compare Meek micrografting 1:2 with Mesh grafting

## 5. Conclusions

Based on the wound and short-term scar outcomes of this study, it is more beneficial to use Mesh grafting as STSG expansion technique than Meek micrografting until the 3-month follow-up for reconstruction of smaller wounds (affected mean TBSA 10 ± 10%). Specifically, Mesh grafting demonstrated higher take rates, shorter time to complete wound healing. Moreover, at the 3-month follow-up, Mesh grafting showed a greater patient preference and satisfaction, improved scar elasticity, had less pigmentation differences, and a better scar quality score. Although Meek micrografting resulted in a significantly smaller donor site size, patients preferred this technique over Mesh grafting at a 1:3 ratio.

## Figures and Tables

**Figure 1 ebj-06-00026-f001:**
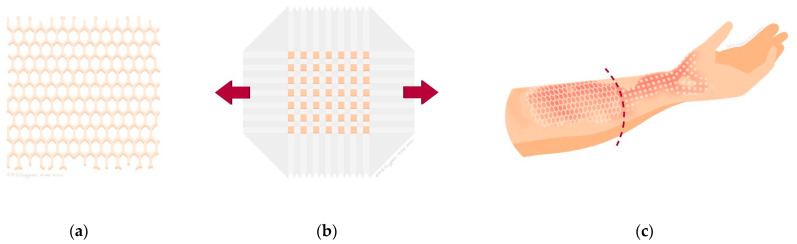
(**a**) Illustration of a Mesh graft; (**b**) illustration of a Meek micrograft on a plissé; (**c**) illustration of a wound divided into two study areas, one covered with a Mesh graft and one with a Meek micrograft.

**Figure 2 ebj-06-00026-f002:**
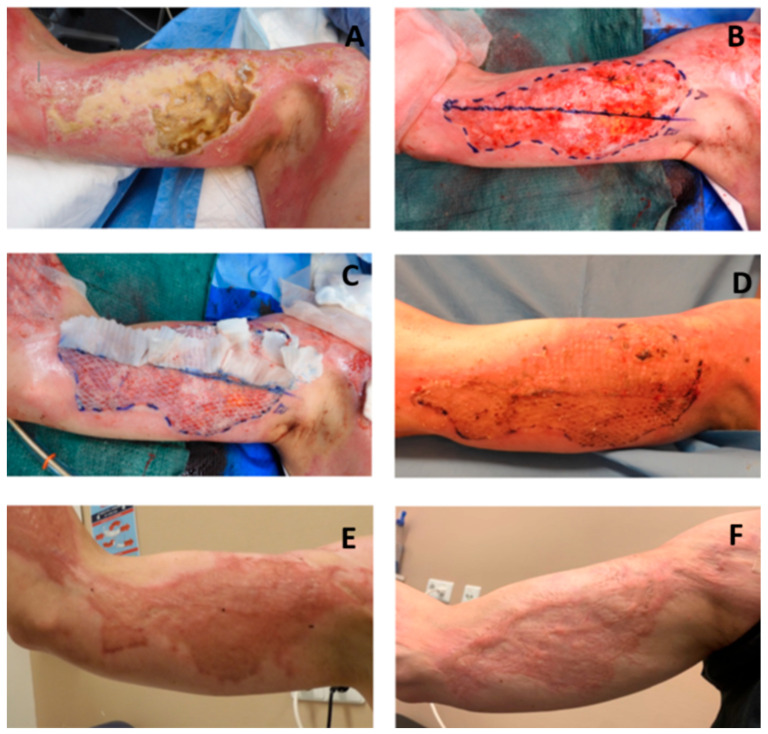
Images of the Meek micrografting and Mesh grafting procedures, wound healing, and scar formation. (**A**) Burn wound on right upper arm; (**B**) after wound debridement and allocation of study area A and B, based on burn depth and size; (**C**) study area A covered with Meek micrografts and B with Mesh graft expansion ratio 1:2; (**D**) take rate evaluation at post-operative day 8; (**E**) scar assessment at 3-month follow-up; and (**F**) scar assessment at 12-month follow-up.

**Figure 3 ebj-06-00026-f003:**
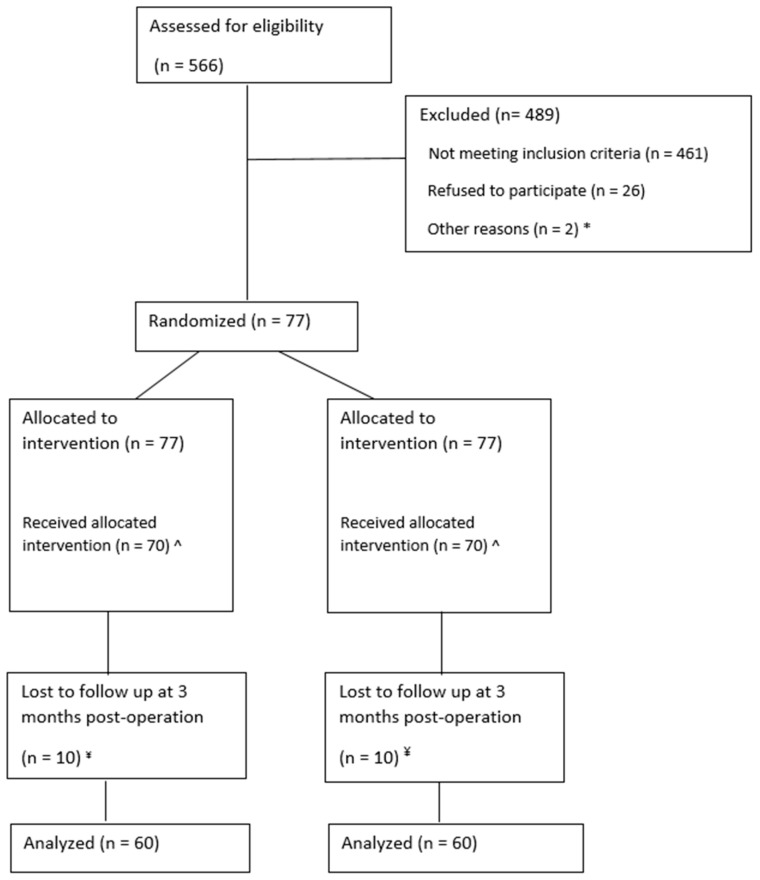
CONSORT diagram. * n = 1 was missed by the research team during a congress abroad. n = 1 patient was transferred to another burn center after screening; ^ n = 3 excluded after allocation because of expansion ratio choice changed to 1:1.5 due to changed medical situation. n = 2 did not receive autografted, treated with ointments only. n = 1 withdrawn after signing informed consent file, but before intervention. n = 1 died from multi-organ failure after allocation, but before intervention; ^¥^ n = 3 died. n = 7 no show at follow-up at 3 months.

**Table 1 ebj-06-00026-t001:** In- and exclusion criteria for study enrollment.

Inclusion Criteria	Exclusion Criteria
Patients ≥ 18 years	Patients who participated in another study utilizing an investigational drug or device within the previous 30 days
Patients with Two comparable deep partial-thickness and/or full-thickness (parts of) burns or skin defects; In case of burns, confirmed with laser Doppler whenever possible; Minimum 36 cm^2^ (=one 1:2 Meek plissé) per study area; Requiring surgery after assessment by a (plastic) surgeon/burn physician	Patients with wounds only covering face, hands or joints
	Patients who had one or more medical condition(s) that in the opinion of the treating physician would make the patient an inappropriate candidate for this study
Patients who are mentally capable to give legal consent or when the patient is temporarily incompetent (e.g., patient is sedated/ventilated), a legal representative who can give legal consent	Patients who are expected (according to the responsible medical doctor) to be non-compliant to the study protocol (this included patients with severe cognitive dysfunction/impairment and severe psychiatric disorders)

**Table 2 ebj-06-00026-t002:** Patient and wound characteristics.

	Value	%
Patients	70	100
Sex (n male)	46	66
Age (years), mean ± SD	58 ± 17	
Diabetes mellitus (n)	14	20
Smoking (n)	9	13
Affected TBSA all wounds (%), mean ± SD	10 ± 10	
Etiology wounds (n)		
Burns	50	71
Flame	35	50
Scald	12	17
Electrical	1	1
Frostbite	1	1
Steam	1	1
Necrotizing soft tissue infection	16	23
Other	4	6
All wound depth (n)		
Deep partial thickness	6	9
Full thickness	21	30
Mix of deep and full thickness	43	61
Expansion ratio study areas (n)		
1:2	60	86
1:3	10	14

Note: TBSA = total body surface area.

**Table 3 ebj-06-00026-t003:** Per- and post-operative wound results.

	Meek	Mesh	*p*-Value *
Study wound size (% TBSA)	1.1 ± 0.8	1.1 ± 0.8	0.602
Study wound size (cm^2^)	148 ± 105	152 ± 123	0.386
Donor site size (cm^2^)	80 ± 60	112 ± 86	**<0.001**
Calculated actual expansion ratio 1:2	2.1 ± 0.7	1.6 ± 0.5	**0.000**
Calculated actual expansion ratio 1:3	2.6 ± 0.7	1.8 ± 0.4	**0.005**
Take at 8 POD (%)	79 ± 25	87 ± 19	**0.003**
Re-epithelialization at 14 POD (%)	80 ± 22	90 ± 13	**0.001**
Complete wound healing study wounds (days)	35 ± 19	30 ± 19	**<0.001**
Complete wound healing donor site (days)	21 ± 11	26 ± 45	0.606
Included patients (n)	70	70	70

* All outcomes are the means ± standard deviation. Mean differences between Meek and Mesh outcomes were tested with Wilcoxon signed-rank tests.

**Table 4 ebj-06-00026-t004:** Patient technique preference at hospital discharge and at the 3-month follow-up.

	Meek Preference	Mesh Preference	No Preference
At hospital discharge			
Total population, n (%)	11 (37)	12 (40)	7 (23)
Included patients (n)	30	30	30
1:2, n (%)	9 (36)	10 (40)	6 (24)
Included patients (n)	25	25	25
1:3, n (%)	2 (40)	2 (40)	1 (20)
Included patients (n)	5	5	5
At 3 months follow-up			
Total population, n (%)	19 (32)	29 (49)	11 (19)
Included patients (n)	59	59	59
1:2, n (%)	13 (26)	27 (54)	10 (20)
Included patients (n)	50	50	50
1:3, n (%)	6 (67)	2 (22)	1 (11)
Included patients (n)	9	9	9

**Table 5 ebj-06-00026-t005:** Crosstab patient satisfaction on Meek micrografting and Mesh grafting at hospital discharge.

		Patient Satisfaction for Mesh					Total
		Very Satisfied	A Little Satisfied	Neutral	A Little Dissatisfied	Very Dissatisfied	
Patient satisfaction for Meek	Very satisfied	**3**	3	2	0	0	8
	A little satisfied	2	**7**	3	0	0	12
	Neutral	2	2	**2**	0	0	6
	A little dissatisfied	1	0	1	**0**	0	2
	Very dissatisfied	0	0	0	0	**1**	1
Total		8	12	8	0	1	29

**Table 6 ebj-06-00026-t006:** Crosstab patient satisfaction for Meek micrografting and Mesh grafting at the 3-month follow-up.

		Patient Satisfaction for Mesh					Total
		Very Satisfied	A Little Satisfied	Neutral	A Little Dissatisfied	Very Dissatisfied	
Patient satisfaction for Meek	Very satisfied	**12**	5	1	0	0	18
	A little satisfied	12	**6**	6	1	0	25
	Neutral	1	6	**1**	1	0	9
	A little dissatisfied	0	1	2	**1**	0	4
	Very dissatisfied	1	0	2	0	**0**	3
Total		26	18	12	3	0	59

**Table 7 ebj-06-00026-t007:** Scar quality results at 3 months measured with the Patient and Observer Assessment Scale 2.0.

	Meek	Mesh	Difference *	95% CI ^¥^	*p*-Value ***
PATIENTS					
Pain	2.3	2.0	0.3	−0.1–0.7	0.126
Itch	3.3	3.1	0.1	−0.5–0.7	0.963
Color	6.2	5.3	1.0	0.3–1.6	**<0.001**
Pliability	5.7	5.1	0.6	0.0–1.2	0.063
Thickness	3.7	3.1	0.6	0.0–1.1	**0.037**
Relief	5.0	3.8	1.1	0.5–1.8	**<0.001**
Overall opinion	5.1	4.3	0.8	0.3–1.4	**0.002**
Included patients	59	59	59	59	59
OBSERVERS					
Vascularity	4.9	4.1	0.8	0.5–1.1	**<0.001**
Pigmentation	3.0	2.9	0.1	−0.2–0.4	0.398
Thickness	2.9	2.6	0.4	0.1–0.6	**0.007**
Relief	3.7	3.2	0.6	0.2–0.9	**0.001**
Pliability	4.2	3.7	0.6	0.3–0.8	**<0.001**
Surface area	3.0	2.7	0.3	0.1–0.5	**0.013**
Overall opinion	4.6	3.9	0.8	0.4–1.1	**<0.001**
Included patients	60	60	60	60	60

* Comparison of the mean Meek and Mesh POSAS results with a Wilcoxon signed-rank test. ^¥^ 95% confidence interval of the mean difference between Meek and Mesh POSAS values.

**Table 8 ebj-06-00026-t008:** Scar pliability results at 3 months post-operation measured with the cutometer.

	Meek	Mesh	*p*-Value ***	Ratio
Uf (mm), mean ± SD	0.52 ± 0.24	0.57 ± 0.26	0.064	0.98 ± 0.32
Ue (mm), mean ± SD	0.37 ± 0.20	0.42 ± 0.21	**0.013**	0.96 ± 0.33
Included patients (n)	58	58	58	58

Note: Scar pliability assessments were missing in 2/60 patients due to logistic reasons. Uf = maximal extension and Ue = elasticity. Ratio was calculated by the mean Meek values/mean Mesh values. * Comparison of the difference between Meek and Mesh scar pliability results with a Wilcoxon signed-rank test.

**Table 9 ebj-06-00026-t009:** Scar color results at 3 months post-operation measured with the dermaspectrometer.

	Meek	Mesh	*p*-Value ***
Erythema, mean difference ± SD	4.9 ± 3.8	5.1 ± 3.5	0.548
Melanin, mean difference ± SD	13.3 ± 8.3	12.1 ± 7.7	**0.019**
Included patients (n)	58	58	58

Note: Scar color assessments were missing in 2/60 patients; 1 due to technical issues and 1 due to logistic reasons. Mean difference scores were calculated using the mean erythema and melanin of Meek and Mesh study areas values minus mean erythema and melanin of patients’ normal skin (same value for Meek and Mesh). * Comparison of difference between Meek and Mesh scar color results with a Wilcoxon signed-rank test.

**Table 10 ebj-06-00026-t010:** Wound colonization results.

	Meek Operation	Mesh Operation	Meek +8 POD	Mesh +8 POD	Meek +14 POD	Mesh +14 POD
Negative (n)	22	24	16	18	9	1
Non-pathogenic flora (n)	28	26	30	25	15	9
*S. aureus* (n)	11	10	11	14	5	6
*P. aeruginosa* (n)	4	4	3	3	4	0
Included patients (n)	65	65	61	61	38	34

Note: *S. aureus* = *Staphylococcus aureus*, *P. aeruginosa* = *Pseudomonas aeruginosa*, *E. coli* = *Escherichia coli*, fungi were not further specified.

**Table 11 ebj-06-00026-t011:** Wound colonization results: non-eventful wound healing for both study areas.

	Meek Operation	Mesh Operation	Meek +8 POD	Mesh +8 POD	Meek +14 POD	Mesh +14 POD
Negative (n)	6	8	6	5	1	1
Non-pathogenic flora (n)	8	6	7	7	4	2
*S. aureus* (n)	1	1	1	1	-	1
*P. aeruginosa* (n)	2	2	1	1	3	3
Included patients (n)	17	17	15	14	8	7

Note: *S. aureus* = *Staphylococcus aureus*, *P. aeruginosa* = *Pseudomonas aeruginosa*, *E. coli* = *Escherichia coli*, fungi were not further specified.

**Table 12 ebj-06-00026-t012:** Wound colonization results: delayed wound healing for both study areas.

	Meek Operation	Mesh Operation	Meek +8 POD	Mesh +8 POD	Meek +14 POD	Mesh +14 POD
Negative (n)	12	12	6	9	10	8
Non-pathogenic flora (n)	12	13	18	14	10	13
*S. aureus* (n)	9	7	9	11	6	4
*P. aeruginosa* (n)	2	2	1	1	1	1
Included patients (n)	35	34	34	35	27	26

Note: *S. aureus* = *Staphylococcus aureus*, *P. aeruginosa* = *Pseudomonas aeruginosa*, *E. coli* = *Escherichia coli*, fungi were not further specified.

**Table 13 ebj-06-00026-t013:** Wound colonization results: delayed wound healing for the Meek study area only.

	Meek Operation	Mesh Operation	Meek +8 POD	Mesh +8 POD	Meek +14 POD	Mesh +14 POD
Negative (n)	3	3	2	3	-	-
Non-pathogenic flora (n)	7	6	5	3	2	-
*S. aureus* (n)	1	2	2	3	1	1
*P. aeruginosa* (n)	-	-	1	1	-	-
Included patients (n)	11	11	10	10	3	1

Note: *S. aureus* = *Staphylococcus aureus*, *P. aeruginosa* = *Pseudomonas aeruginosa*, *E. coli* = *Escherichia coli*, fungi were not further specified.

## Data Availability

The raw data supporting the conclusions of this article will be made available by the authors on request.

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
