# Peer review of "Wound and Short-Term Scar Outcomes of Meek Micrografting Versus Mesh Grafting: An Intra-Patient Randomized Controlled Trial"

_2673-1991, 2025, doi:10.3390/ebj6020026_

Round 1

Reviewer 1 Report

Comments and Suggestions for Authors

In this manuscript, the authors compare two grafting techniques—Meek and Mesh. The topic is of extreme importance to the burn community, as objective data on Meek micrografting, which is growing in popularity are needed. Overall, the study design is solid, especially because both procedures are applied to the same patients, minimizing bias. However, the objective evaluation of scarring could have been more detailed: beyond scar pliability and color, parameters such as dermal thickness or water-binding capacity could also have been measured. This could be added to the limiations section of the manuscript. Likewise there is significant loss to follow up, particularly regarding preference assessment, which should be addressed in limitations. Overall I would suggest to highlight in the discussion that this study features quite small burns and that potential benefits of meek micrografting (such as true expansion, better handling, etc.) might be masked in small expansions and grafting areas and become apparent in analyses of larger burns. This is specifically not meant to take away in any way from this study which is the first of its kind and very important. 

A few additional points need clarification. First, the sequence of presenting Meek versus Mesh results should be consistent throughout the manuscript to avoid confusion; if Mesh is presented first, that order should be maintained. Second, there is a discrepancy between the 477 patients reported as screened in the Results section and the 566 patients shown as assessed in Figure 3. Third, please explain why not all patients were asked about their preference at both time points. Lastly, a clearer graphical comparison—such as bar charts or box plots—would help readers grasp the differences more easily, since Figures 5a and 5b seem confusing in their current form.

Author Response

Reviewer 1

In this manuscript, the authors compare two grafting techniques—Meek and Mesh. The topic is of extreme importance to the burn community, as objective data on Meek micrografting, which is growing in popularity are needed. Overall, the study design is solid, especially because both procedures are applied to the same patients, minimizing bias.
The authors would like to thank the reviewer for the compliment and the feedback on the manuscript!

  1. However, the objective evaluation of scarring could have been more detailed: beyond scar pliability and color, parameters such as dermal thickness or water-binding capacity could also have been measured. This could be added to the limitations section of the manuscript.
    The authors appreciate the reviewer’s valuable suggestion regarding the inclusion of additional objective parameters such as dermal thickness and water-binding capacity in the evaluation of scarring. We fully agree that these are relevant and informative measures. However, in the context of this study, we have already incorporated several objective parameters alongside patient reported outcomes, providing a fairly comprehensive evaluation of scar quality. Furthermore, the scar measurement set used in this study is the standardized measurement protocol employed across most related research projects at our center. Maintaining consistency in measurement methods facilitates comparability between studies and contributes to building a coherent dataset for future analyses. While in the inclusion of additional parameters may offer further insight, we believe the current set sufficiently serves the objectives of this study.
  2. Likewise there is significant loss to follow up, particularly regarding preference assessment, which should be addressed in limitations.
    The authors agree with the reviewer. A 10 % loss to follow-up had been anticipated; however, owing to various factors, this threshold was exceeded to 10 patients at 3 months follow-up. The authors will address this in the limitations section. Moreover, as the reviewer correctly noted, patient preferences at hospital discharge were assessed less frequently; this was attributable to two reasons: first, some patients underwent outpatient surgery and left the hospital the same day as the surgery. These patients were not asked for their preference at hospital discharge. Second, some patients were missed at hospital discharge and therefore could not be asked for their preference.

The authors added this to the discussion section: “A limitation of the present study is that ten patients were lost to follow-up, exceeding the seven patients originally anticipated. Furthermore, patient preferences at the time of hospital discharge were not consistently recorded. This inconsistency arose for two reasons: first, a substantial proportion of patients received care on an outpatient (day-treatment) basis; and second, several patients were inadvertently overlooked at the moment of discharge.”

See comment #rev 1.2

  1. Overall I would suggest to highlight in the discussion that this study features quite small burns and that potential benefits of meek micrografting (such as true expansion, better handling, etc.) might be masked in small expansions and grafting areas and become apparent in analyses of larger burns. This is specifically not meant to take away in any way from this study which is the first of its kind and very important. 
    Thank you for this suggestion. We fully agree that Meek micrografting has demonstrated clear advantages in the treatment of extensive burns. Indeed, it was precisely our positive experience with Meek in largesurface area burns that prompted us to explore its applicability in smaller burns.. Accordingly, this study focuses specifically on comparing Meek and mesh techniques in relatively smaller burns. In response to your comment, we have added the following sentence to the discussion section:

“It should be noted that our cohort comprised relatively small burn areas; consequently, certain advantages of Meek micrografting, such as true graft size and the possibility of higher expansion ratios, enabling a smaller donor site, may be less apparent in small grafting areas but are likely to emerge more clearly in larger burns.”

See comment #rev 1.3

  1. A few additional points need clarification. First, the sequence of presenting Meek versus Mesh results should be consistent throughout the manuscript to avoid confusion; if Mesh is presented first, that order should be maintained.
    The authors thank the reviewer for this suggestion and fully agree with the suggestion. Accordingly, they have revised the Results section so that Meek results are now presented first throughout, followed by Mesh results.

See several sentences in Results section

  1. Second, there is a discrepancy between the 477 patients reported as screened in the Results section and the 566 patients shown as assessed in Figure 3.
    You are right, that is discrepancy based on an typing error. However, the manuscript states 431 patients were screened (sentence number 199), we assume that is what you mean. 431 were the amount of patients from one burn center only.

The authors changed the “431” to “566 patients” in line 199.

see comment #rev 1.5.

  1. Third, please explain why not all patients were asked about their preference at both time points.
    Please see comment #rev 1.2
  2. Lastly, a clearer graphical comparison—such as bar charts or box plots—would help readers grasp the differences more easily, since Figures 5a and 5b seem confusing in their current form.
    The authors would like to thank the reviewer for emphasizing the value of graphical comparisons for clarity. In this particular instance, however, the authors found that the precise counts, and the nuanced overlap and/or divergence in patient preferences between Meek micrografting and mesh grafting, are more immediately apparent in the tabular format. Should the reviewer of the editor nonetheless prefer a visual representation, the authors could convert the crosstabs into stacked-bar or similar charts as exemplified below.

Reviewer 2 Report

Comments and Suggestions for Authors Dear authors, your topic is actual, managing wounds for the doctor is a dynamic and multifactorial process. Therefore, making an overall assessment, for hospitals that do not have other devices for skin extraction, the Mesh graft still has a way to be used even in small wounds for the post-operative outcome. The "materials and methods" part should be explored in order to better explain to "non-experts" what both techniques, Meek and Mesh, consist of in detail.
Overall It is a topic to be explored and on which to do further studies to improve what is very attentive today: Wound Healing.

Author Response

Reviewer 2

Dear authors, your topic is actual, managing wounds for the doctor is a dynamic and multifactorial process. Therefore, making an overall assessment, for hospitals that do not have other devices for skin extraction, the Mesh graft still has a way to be used even in small wounds for the post-operative outcome. The "materials and methods" part should be explored in order to better explain to "non-experts" what both techniques, Meek and Mesh, consist of in detail.
Thank you for highlighting the need to better explain both Meek and Mesh techniques for readers who may be less familiar with them. We acknowledge that the current manuscript presents a compact overview intended primarily for an expert audience. However, a comprehensive description of each grafting method has already been published in our full study protocol (Rijpma et al. 2023). To address your concern, we will add the following sentence to the Materials and Methods section:

“For a detailed description of both Meek and Mesh grafting techniques, including step-by-step procedural guidance, please see the study protocol [15].”

See #rev 2.

Overall It is a topic to be explored and on which to do further studies to improve what is very attentive today: Wound Healing.
The authors would like to thank the reviewer for the compliment and the feedback on the manuscript!

Reviewer 3 Report

Comments and Suggestions for Authors

Dear colleagues!

Comparison of the clinical effectiveness of Meek autologous skin transplantation and mesh dermal transplantation is the subject of numerous studies conducted over the past decades. The question of which of these skin transplantation methods to choose for the treatment of each specific patient is really important for practicing doctors. Each of the methods has a deep pathophysiological justification, mass positive examples of clinical application, and a developed technical instrumental base. The general professional opinion is now inclined to believe that the Meek technology (it is significantly more expensive, requires more time) should be preferred in cases of a sharp deficit of plastic resources - when it is necessary to use the maximum graft expansion ratios - from 1:3 to 1:9. It is important that the authors of the article drew attention to the "overlapping" areas of possible application of Meek and Mesh transplantation - on patients with small wounds, without a deficit of plastic resources. In their treatment, the optimal graft expansion ratio is 1:1.5, 1:2, 1:3. In such a situation, the choice of the autodermoplasty method is not at all obvious; it requires a deep and comprehensive analysis of the consequences. It is necessary to carefully evaluate the wound process in the early postoperative period, study the quality of the restored skin, and study the required area of ​​the donor sites. These parameters were the subject of study by the authors of the article. Such subtle comparisons are most informative when using the “one wound, two treatment methods” design. Studies with such a design are labor-intensive and are conducted relatively rarely. The authors' contribution to solving an urgent and practically significant problem should certainly be welcomed.
At the same time, the article would have become even more interesting and even more valuable if the authors had made some minor adjustments and explanations to it.
1. The authors indicate that “…In the period up to 3-month follow-up, 14/70 (20%) patients were re-operated in one or both study areas. The majority of reoperated patients, 8/14 (57%), were re-autografted at the Meek study area, in 6/14 (43%) patients both study areas were involved and no patients were re-operated for the Mesh study area only…” (lines 249-252). Thus, the majority of patients operated on using the Meek method required a second operation. At the same time, in the early postoperative period, the survival rate of Meek grafts was 79% (line 226). What is the reason for the need for a second operation with a good early transplant result? Did Meek grafts become necrotic in the late period? This clarification should be made either in the description of the results or in the Discussion section.
2. On line 260, the word “relief” is repeated twice. 3. In the "conclusions" section, it would be useful to clarify "Based on wound and short-term scar outcomes of this study, it is more beneficial to use Mesh grafting as STSG expansion technique than Meek micrografting until 3-month follow-up for reconstruction of smaller wounds (affected mean TBSA 10 ± 10%) and when using an expansion ratio of no more than 1:2". Overall, the topic, study design, thoroughness of the work performed by the researchers, and the validity of the conclusions make the most favorable impression. In my opinion, the article can be published after making minimal adjustments.

Author Response

Reviewer 3

Dear colleagues!
Comparison of the clinical effectiveness of Meek autologous skin transplantation and mesh dermal transplantation is the subject of numerous studies conducted over the past decades. The question of which of these skin transplantation methods to choose for the treatment of each specific patient is really important for practicing doctors. Each of the methods has a deep pathophysiological justification, mass positive examples of clinical application, and a developed technical instrumental base. The general professional opinion is now inclined to believe that the Meek technology (it is significantly more expensive, requires more time) should be preferred in cases of a sharp deficit of plastic resources - when it is necessary to use the maximum graft expansion ratios - from 1:3 to 1:9. It is important that the authors of the article drew attention to the "overlapping" areas of possible application of Meek and Mesh transplantation - on patients with small wounds, without a deficit of plastic resources. In their treatment, the optimal graft expansion ratio is 1:1.5, 1:2, 1:3. In such a situation, the choice of the autodermoplasty method is not at all obvious; it requires a deep and comprehensive analysis of the consequences. It is necessary to carefully evaluate the wound process in the early postoperative period, study the quality of the restored skin, and study the required area of ​​the donor sites. These parameters were the subject of study by the authors of the article. Such subtle comparisons are most informative when using the “one wound, two treatment methods” design. Studies with such a design are labor-intensive and are conducted relatively rarely. The authors' contribution to solving an urgent and practically significant problem should certainly be welcomed.
At the same time, the article would have become even more interesting and even more valuable if the authors had made some minor adjustments and explanations to it.
The authors would like to thank the reviewer for the compliment and the feedback on the manuscript!

  1. The authors indicate that “…In the period up to 3-month follow-up, 14/70 (20%) patients were re-operated in one or both study areas. The majority of reoperated patients, 8/14 (57%), were re-autografted at the Meek study area, in 6/14 (43%) patients both study areas were involved and no patients were re-operated for the Mesh study area only…” (lines 249-252). Thus, the majority of patients operated on using the Meek method required a second operation. At the same time, in the early postoperative period, the survival rate of Meek grafts was 79% (line 226). What is the reason for the need for a second operation with a good early transplant result? Did Meek grafts become necrotic in the late period? This clarification should be made either in the description of the results or in the Discussion section.
    Thank you for raising this point, the authors understand how the data as originally presented may cause confusion. The overall mean take rate of 79% (SD 25%) across all 70 patients indeed reflects considerable variability. In the 8 patients who required re-operation in the Meek study area, the mean take rate was substantially lower, at 56% (SD 31%), which directly explains the need for early re-grafting. Meek grafts did not become necrotic in the late follow-up period. To clarify this in the Results section, we have added the following sentence:

“….the mean Meek take rate of these 8 patients was with 56±31% quite lower compared to mean Meek take rate of 79±25% of the total study population. This accounted for the decision to re-graft; no cases of late Meek graft necrosis were observed

See #rev 3.1

  1. On line 260, the word “relief” is repeated twice.
    The authors agree with the reviewer, this was a typing error. The second ‘relief’ should be ‘thickness’ and is replaced.

Line 260 the second ‘relief’ is replaced by ‘thickness’.

  1. In the "conclusions" section, it would be useful to clarify "Based on wound and short-term scar outcomes of this study, it is more beneficial to use Mesh grafting as STSG expansion technique than Meek micrografting until 3-month follow-up for reconstruction of smaller wounds (affected mean TBSA 10 ± 10%) and when using an expansion ratio of no more than 1:2".
    Thank you for this suggestion. Our Conclusions section was intentionally drafted as a compact summary of our key findings, with the separate outcome parameters provided elsewhere in the Result and Discussion section. To enhance clarity without detracting from its brevity, we have revised the Conclusions to list the primary outcome parameters separately. The authors added the following sentences to the Conclusion:

“Specifically, Mesh grafting demonstrated higher take rates, shorter time to complete wound healing. Moreover, at 3 months follow-up Mesh grafting showed a greater patient preference and satisfaction, improved scar elasticity, less pigmentation differences and a better scar quality score.”

See #rev 3.

Overall, the topic, study design, thoroughness of the work performed by the researchers, and the validity of the conclusions make the most favorable impression. In my opinion, the article can be published after making minimal adjustments.
The authors would like to thank the reviewer for the very positive assessment of our work.

Reviewer 4 Report

Comments and Suggestions for Authors

The aforementioned paper addresses the comparison of wound healing and short-term scar outcomes between Meek micrografting and mesh grafting techniques. There are existing studies in the literature that explore the same topic, employing similar or identical methodologies.

The authors have presented the results in a clear and concise manner. The paper is methodologically sound, and the references are appropriate for the subject matter discussed.

However, I believe it would be beneficial to provide a more detailed explanation regarding the criteria for patient exclusion from the study. Specifically, the rationale for excluding patients with burns on the hand, particularly on the dorsal side, should be further clarified.

Author Response

Reviewer 4

The aforementioned paper addresses the comparison of wound healing and short-term scar outcomes between Meek micrografting and mesh grafting techniques. There are existing studies in the literature that explore the same topic, employing similar or identical methodologies. The authors have presented the results in a clear and concise manner. The paper is methodologically sound, and the references are appropriate for the subject matter discussed.
The authors would like to thank the reviewer for the compliment and the feedback on the manuscript!

However, I believe it would be beneficial to provide a more detailed explanation regarding the criteria for patient exclusion from the study. Specifically, the rationale for excluding patients with burns on the hand, particularly on the dorsal side, should be further clarified.
The authors would like to thank you for this suggestion. The authors excluded these locations as potential study areas because these wounds were covered with SSGs with relatively smaller expansion ratios of 1:1 to a maximum of 1:1.5. Accordingly, the authors have added the following sentence to the Methods section:

“…. wounds covering only the face, hands, or joints were excluded because these locations are covered with STSGs with a maximum expansion ratio of 1:1.5, which is not achievable with Meek micrografting and are thus unsuitable as study areas.”

See #rev 4.

Round 2

Reviewer 1 Report

Comments and Suggestions for Authors

my concerns have been addressed sufficiently and the paper is ready for publication

Reviewer 3 Report

Comments and Suggestions for Authors

Dear colleagues,

I have no more fundamental comments. Interesting topic, well-done research, excellent article. I wish you every success.

Reviewer 4 Report

Comments and Suggestions for Authors

Thank you for the corrections. The paper can be accept for publication.